# Inventory and GLOF Susceptibility of Glacial Lakes in Hunza River Basin, Western Karakorum

Fakhra Muneeb [1], Siddique Ullah Baig [2], Junaid Aziz Khan [3] and Muhammad Fahim Khokhar [1,*]

1 Institute of Environmental Sciences and Engineering, National University of Sciences and Technology, Islamabad 44000, Pakistan; fmuneeb.mses2018iese@student.nust.edu.pk
2 High Mountain Research Center, Department of Development Studies COMSATS University Islamabad, Abbottabad 22060, Pakistan; sbbaig@cuiatd.edu.pk
3 Institute of Geographical Information System, National University of Sciences and Technology, Islamabad 44000, Pakistan; junaid@igis.nust.edu.pk
* Correspondence: fahim.khokhar@iese.nust.edu.pk; Tel.: +92-51-90854308

**Abstract:** Northern latitudes of Pakistan are warming at faster rate as compared to the rest of the country. It has induced irregular and sudden glacier fluctuations leading to the progression of glacial lakes, and thus enhancing the risk of Glacier Lake Outbursts Floods (GLOF) in the mountain systems of Pakistan. Lack of up-to-date inventory, classification, and susceptibility profiles of glacier lakes and newly formed GLOFs, are few factors which pose huge hindrance towards disaster preparedness and risk reduction strategies in Pakistan. This study aims to bridge the existing gap in data and knowledge by exploiting satellite observations, and efforts are made to compile and update glacier lake inventories. GLOF susceptibility assessment is evaluated by using Analytical Hierarchy Process (AHP), a multicriteria structured technique based on three susceptibility contributing factors: Geographic, topographic, and climatic. A total of 294 glacial lakes are delineated with a total area of $7.85 \pm 0.31$ km$^2$ for the year 2018. Analysis has identified six glacier lakes as potential GLOF and met the pre-established criteria of damaging GLOFs. The historical background of earlier GLOF events is utilized to validate the anticipated approach and found this method appropriate for first order detection and prioritization of potential GLOFs in Northern Pakistan.

**Keywords:** glacial lake; outbursts susceptibility; GLOF; analytical hierarchy process; Hunza River Basin

## 1. Introduction

Climate change is the main driving force in evolution and growth of glacial lakes. The average temperature of Pakistan has been increased by 1.04 °C with the rate of 0.09 °C from 1960 to 2014 [1]. According to Pakistan Meteorological Department data, the mean temperature in Northern parts of Pakistan was increased by 0.8 °C during 1900–2000 while 0.6 °C was recorded in Southern parts during the same time period [2]. The higher level of temperature extremes is alarming, and it is projected under the RCP 8.5 scenario by the end of 21st century to increase 4.8 °C in Northern Pakistan as compared to other regions, i.e., Khyber Pakthunkhwa with increase of 4.6 °C and Monsoon areas with 4.5 °C increase. This makes the Northern Pakistan a more vulnerable region to regional temperature fluctuations and causes frequent climatic extreme events to occur [3]. Especially in winters, maximum temperature has been enhanced by 1.79 °C in Upper Indus Basin from 1967 to 2005 and an annual change rate of 0.04 °C was identified. [4]. Glacier retreat owing to increasing temperature has led to an increasing number and spatial extent of glacial lakes in high mountains worldwide [5–8]. Especially, the glacial ice in the Hunza River Basin has been declining from 44.02% recorded in 1989 to 34.99% by the year 2010 [9]. Over the last decade, escalating temperature is instigating glacier recession, directing to the evolution of glacial lakes in Hunza River Basin [10].

Glacier Lake Outbursts Floods (GLOFs) pose serious threats in high mountain glaciated environments since the last decade, owing to a temperature warming rate accompanied by deglaciation [11,12]. Furthermore, overwhelmingly increasing population, and anthropogenic and socio-economic developmental activities have exacerbated the destabilization of fragile mountain areas, making them more susceptible to GLOF events and causing severe damages [8,13]. Because of the ever-increasing average temperature and consequent glacier retreat, the intensity and frequency of GLOF events are likely to enhance considerably in the future [14]. Approximately 3044 glacial lakes have been identified in HKH region of Pakistan, among which about 36 lakes are recognized as critical GLOF hazards [15,16]. The probability of GLOF occurrence cannot be estimated through standardized statistical methods [17]. Moreover, it is quite challenging to estimate GLOF events due to complexity involved in external or internal triggering mechanisms, low occurrence of GLOF events, and dynamic fluctuations in glacial systems [18]. In high mountain regions where ground data collection and field observations are obstructed by harsh weather and climatic conditions, remote sensing techniques offer flexible approaches for spatial and temporal assessment and monitoring of glacial lakes and potential GLOFs [19–22].

Earlier studies have reported that glaciers in the Hunza River Basin are advancing or at least in balance since the 1970s [23,24]. However, for the last few years, a series of rapidly emerging tendencies of creating glacier lakes is reported in Shimshal and Passu valleys in the Hunza River Basin area. Climate change-induced glacier fluctuations led to the formation and growth of glacial lakes, enhancing the risk of GLOFs. Especially, climate change-induced glacier fluctuations are important for management of water resources, assessment of associated hazard potential, and estimation of future spatio-temporal growth of glacier lakes [25–27]. Thus, there is a dire need to focus on up to date knowledge and understanding about glacier lakes and their analysis and extent for management of water, assessing potential GLOF hazard and risk management. An outdated inventory, classification, and susceptibility profiles of newly formed GLOFs is available. This study aims to provide an up-to-date inventory of glacial lakes in the Hunza River Basin, Western Karakorum, by using Sentinel 2 imagery and to assess their GLOF susceptibility. A multicriteria Analytical Hierarchy Process (AHP) method is used to assess the magnitude (e.g., low, medium, and high) of glacial lakes susceptibilities. AHP based multi-criteria has been widely employed in other regions for outbursts susceptibility assessment [28–33]. Similar techniques for examining the outbursts susceptibility were used in Uzbekistan [34], Cordillera blanca, Peru [18], and the Himalayan region [35]. These methods are well suitable for Northern Pakistan's Hunza River Basin. The historical background of earlier GLOF events is utilized to validate the anticipated approach. Monitoring of hotspots of potential GLOFs are recommended to be reinforced and comprehend the GLOF events to reduce its adverse effects. In the current study, various parameters were chosen based on three reasons. Foremost, these indicators are widely utilized for assessing GLOF outburst susceptibility. Secondly, these factors are easily interpreted and measured through remote sensing data, and field work is not required. Third, indicators being nominal and continuous data could be used as semi quantitative or qualitative for evaluating outbursts susceptibility.

## 2. Study Area

This study was carried out in Hunza River Basin (HRB), which lies within the Karakoram Range. The spatial domain of the study area is 36°32′N–37°05′N, 74°02′E–75°48′E (Figure 1). The study area is renowned for: (1) A concentration of high-pitched mountains and glaciers; (2) higher mean elevation as compared to other landscapes of the Karakoram Range; (3) entails both: Debris-covered glaciers at low elevation and clean glaciers at high elevation. Elevation ranges from 7850 m above sea level (m.a.s.l.) of Batura glacier (in Upper Hunza) to 1395 m at Danyor suspension bridge. Geographically, HRB is a huge mountain landscape extending from the border of Xinjiang province of China up to the Wakhan Corridor of north-east Afghanistan.

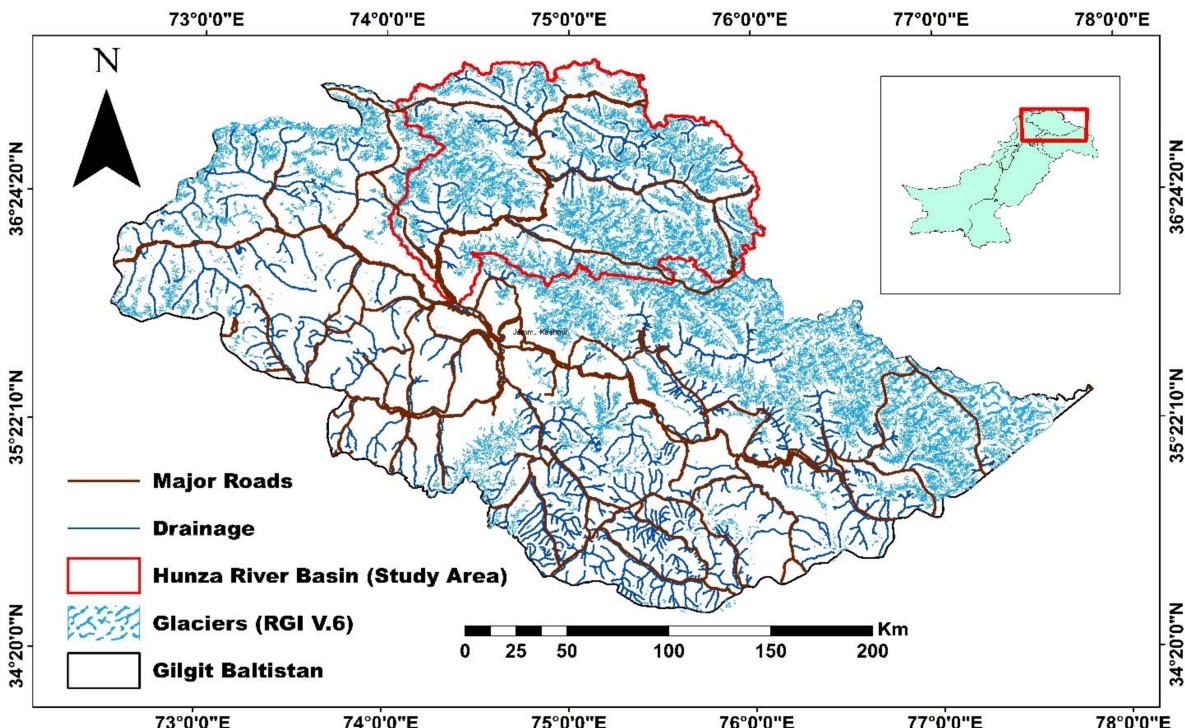

**Figure 1.** Map of the study area (Hunza River Basin), lies in the Northern Pakistan, encompassing the ranges (Karakoram, Hindu-Kush, and Himalaya) occupying 13757 Karakoram glaciers, major roads/rivers, and the Indus River passing through the Karakoram/Himalayan mountains originating in the People's Republic of China.

### 2.1. Bedrock Geology and Tectonics

The stratigraphy of Northern Karakoram is suitable for comprehensive study and suggests a complex geography of the area during the Paleozoic and Mesozoic eras [36]. The Reshun Fault in Chitral ties with the Upper Hunza Fault 200 km to the east, exhibiting continuity with the tectonic structures in the upper Chupurson valley. In Chitral, large thrust sheets consist of wide-ranging Paleozoic to Mesozoic successions, whereas in Hunza, thrust sheets are comprised of Permian to Mesozoic successions. The Karakoram Batholith is a large body of intrusive rock in the region. A transect from south to north indicates the Hunza Plutonic Complex crossing over into the Batura Plutonic Complex to the north, which consists of granites and granodiorites [37]. In the north, bimodal plutons, Mg-K met aluminous granitoids with biotite and amphibole, and two-mica peraluminous granitoids are present. Furthermore, to the west of Hunza, along the Karamber transect, is found the Hunza Plutonic Complex in the south, which is non-alkaline, and the subalkaline porphyritic granite to the north [36].

### 2.2. Geography and Glaciers

The Hunza River Basin covers a 13,730 km$^2$ area. About ~30% land is covered by ~1300 glaciers. These glaciers and their tributaries provide melt-water to the Hunza River, passing through Khunjerab valley, the highest altitude region of the Hunza River Basin adjacent to Kashgar of China. At the middle altitude, two main streams (e.g., Misgar and Chupurson) of Ziarat contributes melt-water to Hunza River followed by Shimshal stream. At the lowest altitude, tributaries of the Hisper glacier also provide melt-water into the Hunza River. All this melt-water is used by the downstream population of Pakistan for irrigation, energy production, and drinking. Details pertaining to salient features of landscapes covering glaciers and barren land of the HRB (the study area) are presented in Table 1.

**Table 1.** Geographic specifications of the study area (Hunza River Basin).

| Characteristics | Descriptions |
|---|---|
| Latitude | 35°54′00″N–37°05′00″N |
| Longitude | 74°02′00″E–75°48′00″E |
| Total basin area | 13,730 km$^2$ |
| | 13,734 km$^2$ |
| Lowest elevation | 1461 m |
| Highest elevation | 7850 m |
| Mean elevation | 5038.73 m |
| Standard dev. | 811.17 |
| Glacier coverage (%) | 28.29% (Calculated from RGI–V.6.0 dataset in this study 30%–39%° |
| Total glacier area | 3886 km$^2$ (calculated from RGI–V.6.0 dataset in this study) |
| No. of glaciers | 1352 (extracted from RGI-V.6.0 dataset in this study)data |

## 3. Materials and Methods

### 3.1. Satellite Data

Satellite remote sensing data is the only accessible way to explore in detail the spatial dynamics of glaciers and glacial lake in the highly inaccessible regions of Northern Pakistan. Therefore, Sentinel 2B imageries with resolution of 10 m were obtained from Earth Explorer (https://earthexplorer.usgs.gov/ accessed on 2 April 2021) to identify and map the glacial lakes in the Hunza River Basin. The imageries were selected to ensure minimal snow and cloud cover. In addition to the above, digital elevation models (DEMs) were obtained from Shuttle Radar Topography Mission (SRTM-DEM) with a spatial resolution of 30 m. It was employed to derive topographic information such as slope, elevation, and aspect of glacial lakes. Sentinel satellite data is used to estimate and update the Randolph Glacier Inventory version (RGI-v.6) glaciers and glacier lake boundary, distance to glacier, and facades variations. Google Earth imageries are also utilized to validate facade variations of glacial lakes synchronized with visual interpretation. Temperature data was also obtained from AIRS with a spatial resolution of 1° × 1°, and precipitation data with a resolution of 0.25° × 0.25° was attained from TRMM and can be accessed from Giovanni (https://giovanni.gsfc.nasa.gov/, last accessed on 30 March 2021) to compute the change in temperature and precipitation over study area.

### 3.2. Glacial Lake Inventory

The step-by-step systematic approach adopted to create the glacial lake inventory is presented in Figure 2. Various scientists and experts have stated automated and semi-automated methods for delineation of glacial lakes [38]. In this study, glacial lakes were identified and extracted from satellite imageries through semi-automated processing techniques accompanied by visual interpretation, comparison with inventory from International Centre for Integrated Mountain Development (ICIMOD) and verification through Google Earth imageries. Glacial lake pixels were automatically mapped using Normalized Difference Water Index (NDWI) as proposed by Huggel et al. [27] according to Equation (1) then manually applied thresholds for extraction of water pixels.

Water bodies have maximum reflection in visible spectra and minimum reflection in near infrared band. Hence, NDWI was estimated using high reflection of green band and intense absorption of NIR band [39,40].

$$NDWI = (Green - NIR)/(Green + NIR) \tag{1}$$

Many researchers have documented a threshold of 0.15 for determining glacial lake areas [38,41]. Hence, this value also serves as a better threshold for lake area identification in the current study.

Manual digitization, editing, and delineation of glacial lakes were also performed through false color composites using near infrared, red, and green bands (8,4,3) to eradicate errors owing to automatic extraction. ArcMap was complemented with Google Earth

for interpretation, examination, and verification of glacial lake boundaries, as it ensures accurate confirmation about the type of lake and drainage of lake.

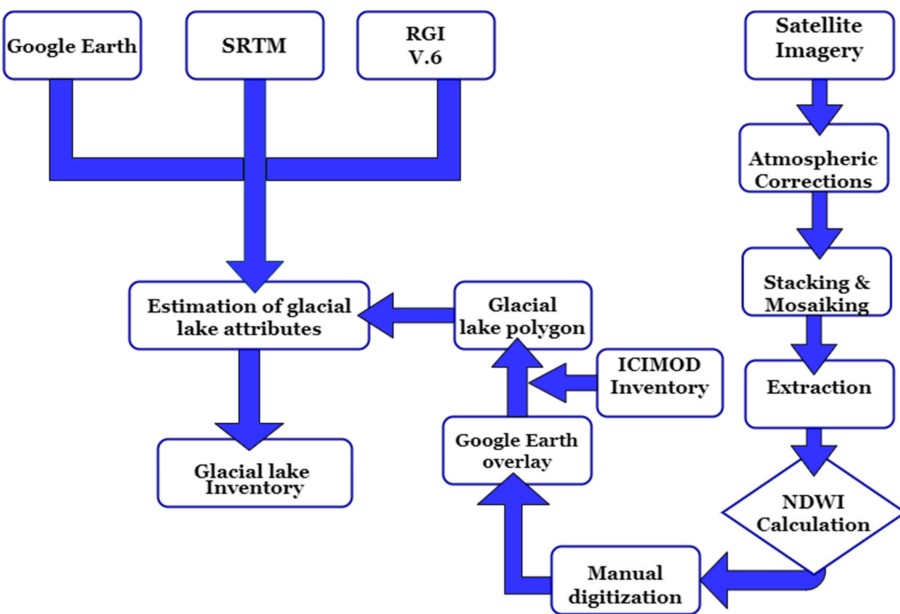

**Figure 2.** Flowchart illustrating the methodology opted for updating of glacial lake inventory.

Area of glacial lakes greater than 0.0008 km$^2$ were catalogued and digitized in the current inventory with the following sets of features dispensed to each lake:

1. Lake ID, where we assigned numbers such as Lake 1 to Lake 294;
2. location of lake in longitude and latitude (WGS 1984);
3. elevation of lake (m);
4. lake area in km$^2$;
5. type of lake: Based on damming material, mapped glacial lakes were categorized as proposed by ICIMOD (2011) [32] by visual interpretation with Google Earth imageries.
   - Moraine dammed lakes are waterbodies impounded by moraine.
   - Ice dammed lakes are waterbodies impounded by ice, including the lakes that form on the surface of a glacier.
   - Bedrock dammed lakes are formed in the depressions eroded and impounded by flat solid rock.
   - Other glacial lakes included landslide lakes and are fed by snow and glacial melt, but damming material is not part of glacial process;
6. distance of glacial lake from glacier in meters;
7. aspect of lake;
8. type: Lakes were also categorized into glacier fed or non-glacier fed on the basis of hydrological connection with glacial water sheds;
9. lake drainage type: Lakes were classified into closed lakes or drained lakes based on outflow of water examined primarily from Google Earth.

### 3.3. Criteria for GLOF Susceptibility

Numerous studies have proposed structures or criteria to identify potential dangerous glacial lakes or to examine consequences of GLOF events [42]. All the parameters (area, dam type, dam characteristics, glacier lake distance, and meteorological conditions) were analyzed and assessed to examine relative importance grounded on information gathered from past GLOF events, literature review, and published reports [14,31]. In this study, we utilized various parameters introduced by different authors for GLOF susceptibility. Various parameters are summarized in Table 2 and Figure 3.

**Table 2.** Index values of parameters with sub criteria.

| Parameters | Critical Values | Index Values (Ci) | Method/Source | References |
|---|---|---|---|---|
| Area of the lake | >0.1 km$^2$ High<br>0.02–0.1 km$^2$ Medium<br>0.01–0.02 km$^2$ low | 1<br>0.5<br>0.25 | Satellite Imagery | Aggarwal et al., 2017 and<br>Bolch et al., 2011 |
| Volume of the lake | $10 \times 10^6$ m$^3$–$100 \times 10^6$ m$^3$ High<br>$1 \times 10^6$ m$^3$–$10 \times 10^6$ m$^3$ Medium<br><$1 \times 10^6$ m$^3$ low | 1<br>0.5<br>0.25 | Empirical formula | Kougkoulos et al., 2018 |
| Type of the lake | Moraine dammed lake High<br>Ice dammed lake Medium<br>Bedrock dammed lake and<br>Other type lake low | 1<br>0.5<br>0.25 | Google Earth | ICIMOD, 2011 |
| Free board level | <5 m High<br>5–15 m Medium<br>Medium >15 m low | 1<br>0.5<br>0.25 | Google Earth/SRTM | Emmer and Vilimek, 2013<br>and Worni et al., 2013 |
| Moraine width to height ratio | <1 High<br>1.0–2.0 Medium<br>>2 low | 1<br>0.5<br>0.25 | Google Earth/SRTM | Wang et al., 2013 |
| Aspect | SE, S, SW<br>N, NE, NW, E, WE, W | 1<br>0.5 | SRTM | Huggel et al., 2002 |
| Drainage | Closed,<br>Open | 1<br>0.5 | Google Earth/<br>Satellite Imagery | Huggel et al., 2002 |
| Distance from Glacier | <80 m High<br>80–600 m Medium<br>>600 m low | 1<br>0.5<br>0.25 | RGI v.6/Satellite Imagery | Wang et al., 2011 |
| Slope of the lake | >20° High<br>3 to 20° Medium<br><3° low | 1<br>0.5<br>0.25 | SRTM | Wang et al., 2013 |
| Lake Growth per Decade (2008–2018) | >100% High<br>50–100% Medium<br>>50% low | 1<br>0.5<br>0.25 | Satellite Imagery | Bolch et al., 2011 |
| Extreme Meteorological events | Frequent: High<br>Sporadic: Medium<br>Unlikely: Low | 1<br>0.5<br>0.25 | AIRS/TRMM | Huggel et al., 2004 |

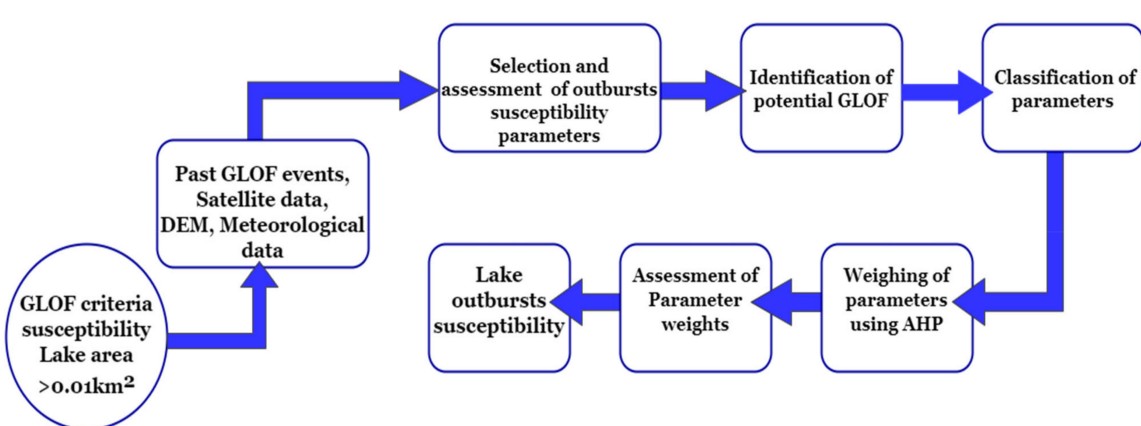

**Figure 3.** Flowsheet illustrating the methodology followed for GLOF susceptibility scheme.

### 3.4. Lake Volume Estimation

A method based on an empirical relationship between depth and area is established to estimate the volume of glacier lakes. The method based on a relationship between lake and volume (Mean lake depth = 0.104 * lake-area$^{1.42}$) for moraine and ice-dammed lakes produced by Huggel et al. [27] is considered to be appropriate to calculate lake-volume by several studies [33,43–46].

Volume of glacial lake in m$^3$ can be expressed as function of the area A (in m$^2$) using the following relationship Equation (2) from Huggel et al. [27]:

$$V = 0.104A^{1.42} \tag{2}$$

### 3.5. GLOF Susceptibility Assessment Using AHP

In the current study, Analytical Hierarchy Process (AHP) was used to prioritize the susceptibility of potential dangerous glacial lakes. AHP is multicriteria decision analysis method for analyzing complex problems entailing various factors. A pairwise comparison matrix assesses the significance of the contributing factor to outbursts by appraising the relative importance of one factor over another. The pairwise comparison matrix is created by allocating values from equal, moderate, strong, very strong, and extreme importance on a scale from 1 to 9. Consistency ratio (CR) was computed on eigen values of factors (Table 3). The measured CR is revised when it exceeds the threshold value of 0.1. We established pairwise comparison matrix by allocating high values to those parameters which are highly associated and contributing more to GLOF. Weights were computed for each parameter. Each variable was further categorized into classes, and final weights were calculated by multiplying class index value with parameter weight. Final weight of lake was computed by adding weights of parameters.

**Table 3.** Pairwise comparison matrix of GLOF susceptibility assessment using Analytical Hierarchy Process (AHP).

| Criteria | Area | Volume | Type | Freeboard Level | W/H Ratio | Aspect | Drainage | Distance from Glacier | Slope | Lake Growth | Climate Variables |
|---|---|---|---|---|---|---|---|---|---|---|---|
| **Area** | 1 | 0.50 | 3.00 | 0.33 | 3.00 | 3.00 | 0.50 | 0.50 | 1.00 | 0.20 | 0.33 |
| **Volume** | 2 | 1 | 3.00 | 0.5 | 1.00 | 2 | 1.00 | 0.33 | 1.00 | 0.33 | 0.33 |
| **Type** | 0.33 | 0.33 | 1 | 0.33 | 2.00 | 2 | 0.50 | 0.33 | 0.33 | 0.33 | 0.20 |
| **Freeboard level** | 3 | 2.00 | 3.00 | 1 | 3.00 | 5.00 | 1.00 | 1.00 | 2.00 | 0.33 | 0.20 |
| **W/H ratio** | 0.33 | 1 | 0.50 | 0.33 | 1 | 2.00 | 0.50 | 1.00 | 0.33 | 0.33 | 0.33 |
| **Aspect** | 0.33 | 0.50 | 0.50 | 0.20 | 0.50 | 1 | 0.50 | 0.33 | 0.33 | 0.33 | 0.20 |
| **Drainage** | 2 | 1 | 2.00 | 1.00 | 2.00 | 2.00 | 1 | 0.33 | 1.00 | 0.33 | 0.33 |
| **Distance from glacier** | 2 | 3.00 | 3.00 | 1.00 | 1.00 | 3.00 | 3.00 | 1 | 0.50 | 0.33 | 0.33 |
| **Slope** | 1 | 1.00 | 3.00 | 0.50 | 3.00 | 3.00 | 1.00 | 2.00 | 1 | 0.20 | 0.33 |
| **Lake growth** | 5 | 3.00 | 3.00 | 3.00 | 3.00 | 3.00 | 3.00 | 3.00 | 5.00 | 1 | 1.00 |
| **Climate variables** | 3 | 3.00 | 5.00 | 5.00 | 3.00 | 5.00 | 3.00 | 3.00 | 3.00 | 1.00 | 1 |

### 3.6. Uncertainty Estimation

The accuracy of the delineated lake is primarily reliant on snow and cloud cover, pixel resolution, preprocessing of image, and the experience and knowledge of the expert [47]. The inaccuracies in glacial lake area delineation are problematic to assess owing to scarcity of field observations. The uncertainty in glacial lake area mapping estimated in this study is based on the approach used by Hanshaw and Bookhagen [48]. It states that errors in lake boundary extraction are Gaussian distributed; 68% of pixels are contingent to errors. The uncertainty in lake boundary is calculated using the following formula in Equation (3):

$$\text{Error}\ (1\partial) = (P/G) \times 0.68 \times G^2/2 \tag{3}$$

where P exhibits the perimeter of the lake, and G denotes grid cell size.

## 4. Results

### 4.1. Glacial Lake Inventory

The current study identified and mapped 294 glacial lakes for the year 2018 in the Hunza River Basin by using Sentinel 2B imageries as shown in Figure 4. When compared to the current study, a roughly 108.51% increase in number and 164.31% increase in glacial lake area was found with respective to 141 glacial lakes having an area of 2.97 km$^2$ delineated by ICIMOD for the year 2005, as shown in Figure 5. The total area of glacial lakes delineated was 7.85 ± 0.31 km$^2$. The detected glacial lakes size ranges from 0.0008 ± 0.0005 km$^2$ to 5.01 ± 0.07 km$^2$.

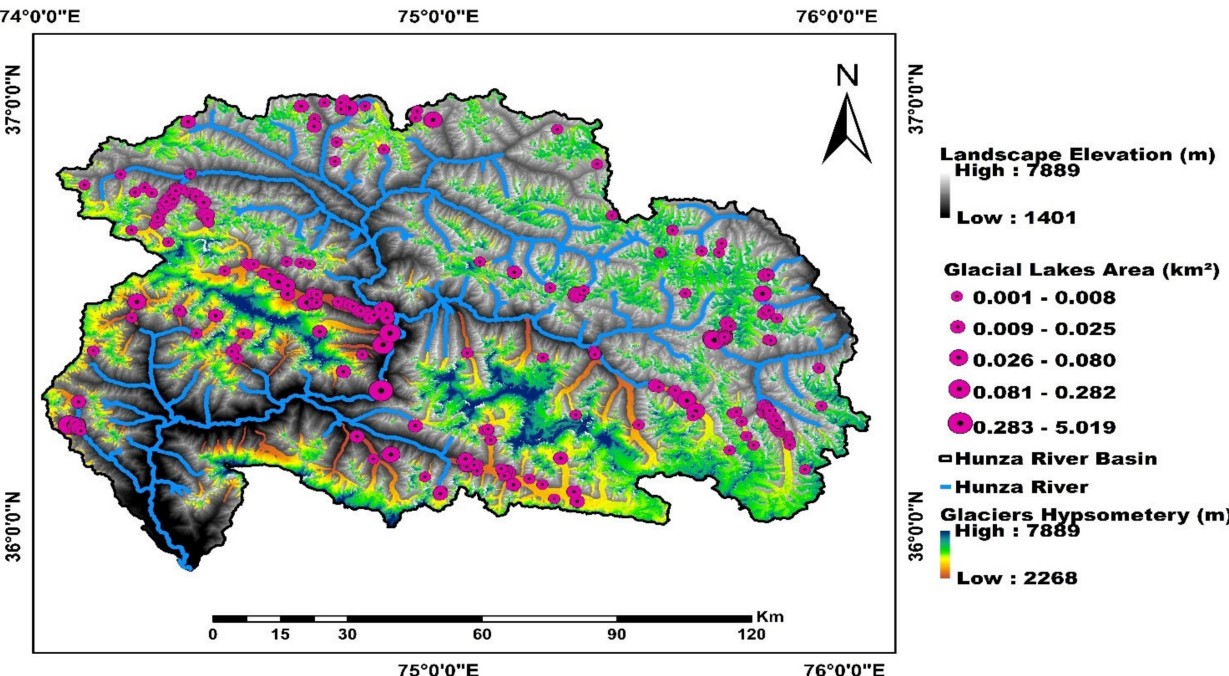

**Figure 4.** The distribution of number and area of each glacier lakes in the Hunza River Basin for the year 2018.

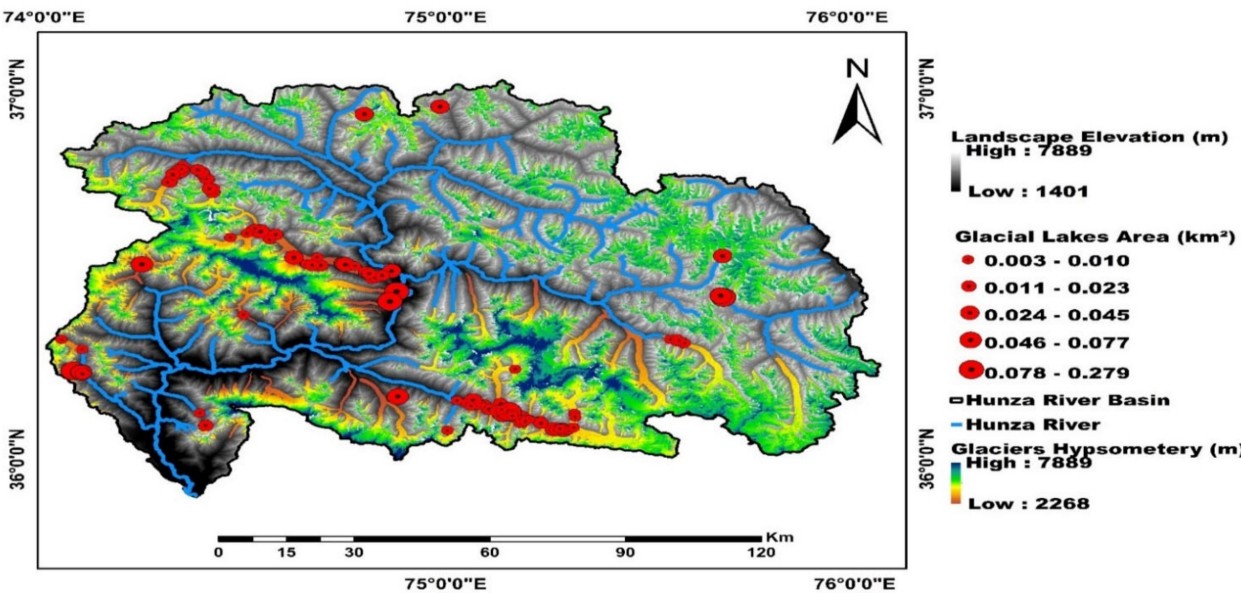

**Figure 5.** The distribution of number and area of each glacier lakes in the Hunza River Basin for the year 2005.

The greater number of glacial lakes in HRB have a smaller size (area < 0.01), encompassing 249 lakes, and accounting for 84.69% of the total number. About 36 glacial lakes have an intermediate area (0.01 $\leq$ area $\leq$ 0.05 km$^2$), and 8 lakes have a large areal extent (0.05 $\leq$ area $\leq$ 0.5 km$^2$), while one lake has a much greater area (area $\geq$ 0.5 km$^2$) (Figure 6a). The existence and size of mapped glacial lakes in the Hunza River Basin exhibited inconsistent distribution features with respective to elevation. The number of glacial lakes indicated an increasing inclination with rising elevation. Glacial lakes situated above 4500 m constituted 28.23% of the total lakes, whereas the greatest area of glacial lakes was located between 2000 to 2500 m owing to a large area of Attabad lake, 5.01 km$^2$, and comprised 63.8% of the total area of glacial lakes (Figure 6b).

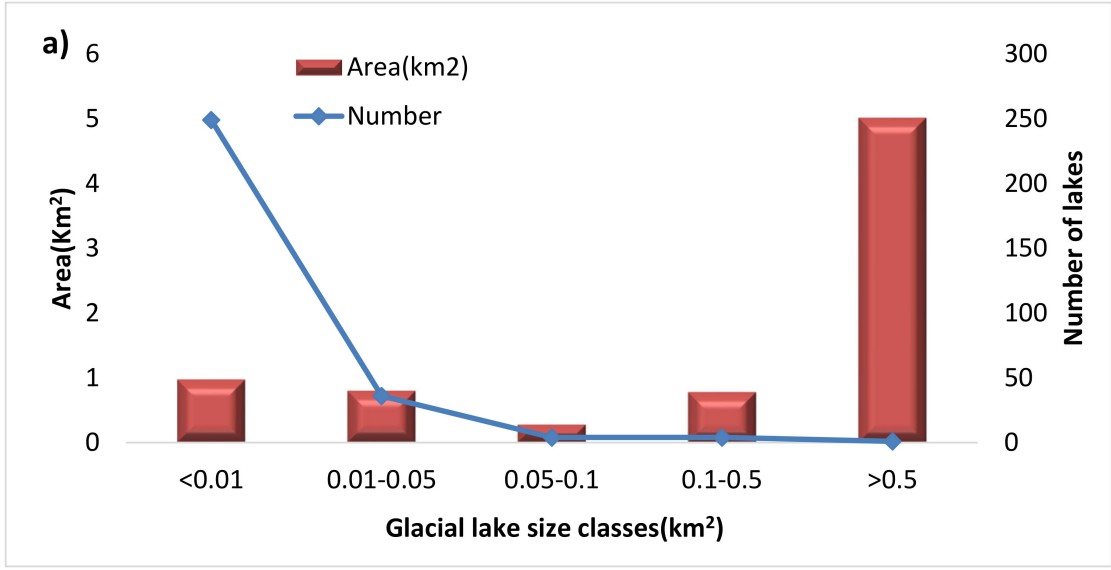

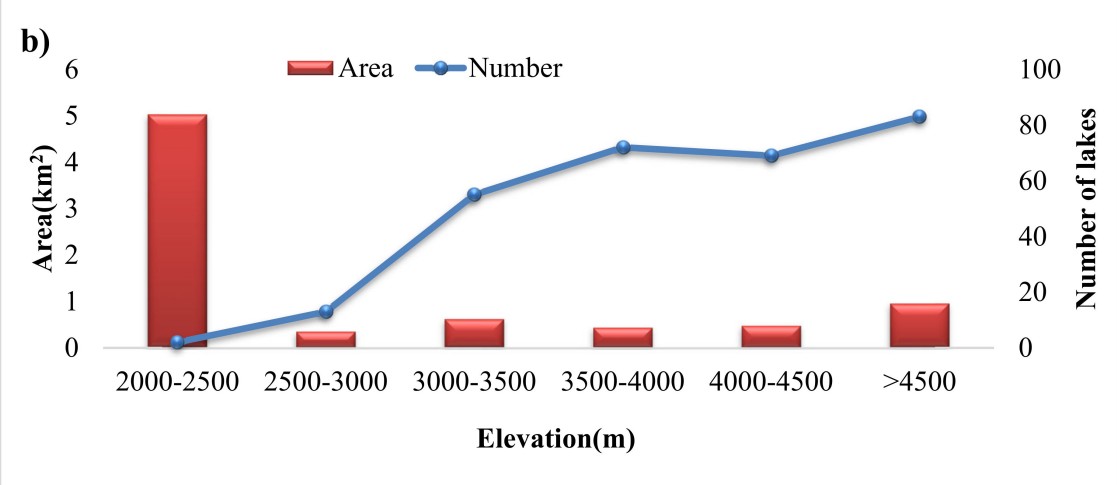

**Figure 6.** Distribution of glacial lakes in different area classes and elevation intervals in the Hunza River Basin in (**a**) number and area of glacial lakes in various glacial lake area classes; (**b**) with different elevation intervals.

Of the 294 glacial lakes, ice dammed lakes are the dominant type, comprising 154 lakes; 52.38% of the total number of glacial lakes. 117 lakes are moraine dammed lakes that account for 39.79 %, followed by 17 Bedrock dammed lakes (5.78%). Six lakes (2.04%) of the total glacial lakes are categorized as other types of glacial lakes, which encompasses landslide blocked lakes and alluvial lakes covering an area of 11.75%, 16.28%, 6.68%, and 65.28%, respectively. The majority of glacial lakes are glacier fed (97.3%) and 2.7 % are non-glacier fed. Surface drainage is explicitly observed in 95.23% lakes, and 4.77% are

closed lakes. The highest number of glacial lakes (18.7%) have SW orientation followed by S (18.02%) then SE (16.32%). Almost 11.9% of total glacial lakes have E aspect. The least number of glacial lakes (5.78%) have N orientation. The highest number of lakes (53.06%) are in contact with the glacier in the Hunza River Basin. About 29.9% lakes of the total glacial lakes lie within 500 m from the glacier. Almost 5.78% glacial lakes lie within the range from 500 m to >1000 m, followed by 5.44% of the total glacial lakes lie from 100 m to >1500 m from the glacier in the basin. About 1.36% of glacial lakes lie from 1500 m to >2500 m from the glacier, and 3.06% lie beyond 2500 m.

Glacial lakes are located at the elevation zone of 2367 m–5264 m. Study revealed that the majority of glacial lakes, about 28.23% of the total glacial lakes, are located at higher elevation of >4500 m. 24.48% are situated in the range 3500 to 4000 m elevation zone, followed by 23.46% lakes are at 4000–4500 m, and only 4.42% of total glacial lakes are in elevation range 2500–3000 m. Moraine dammed lakes are dominated at higher elevations (>4500 m) with mean elevation of 4298 m, whereas ice dammed lakes are situated at medium elevation ranges from 3500 to 4000 m with mean elevation of 3871 m. Bedrock dammed lakes are found at an elevation range of 3000–5000 m with a mean elevation of 4080 m, and other type glacial lakes are dominant at lower elevation varying from 2000–3500 m, having a mean elevation of 2969 m (Figure 7).

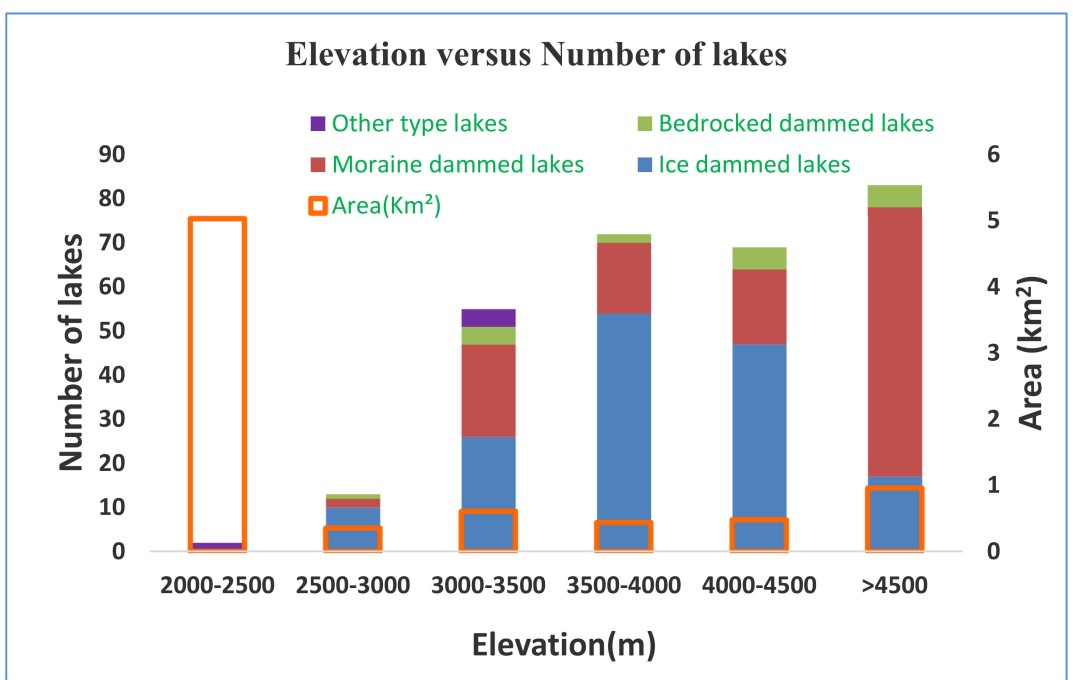

**Figure 7.** Distribution of type of glacial lakes and area with respect to elevation.

### 4.2. Identification of Lakes as Potential GLOFs

The main purpose was to identify glacial lakes as potential GLOFs that pose serious threats of loss of human lives, infrastructure, and to help in development of rational and scientific disaster reduction strategies. Utilizing this new inventory, the potential for GLOF is scrutinized on the basis of multicriteria decision analysis. Glacial lakes with an areal extent of >0.01 km$^2$ (n = 45) have been considered for GLOF susceptibility. About 6 out of 45 glacial lakes identified in the Hunza River Basin met the pre-established criteria that have the potential to be the source of damaging GLOFs as shown in Figure 8. The identified 6 glacial lakes were evaluated to examine the degree of GLOF susceptibility using the AHP. The weights for the designated parameters were computed by means of pairwise comparison method using AHP, and are tabulated in Table 4.

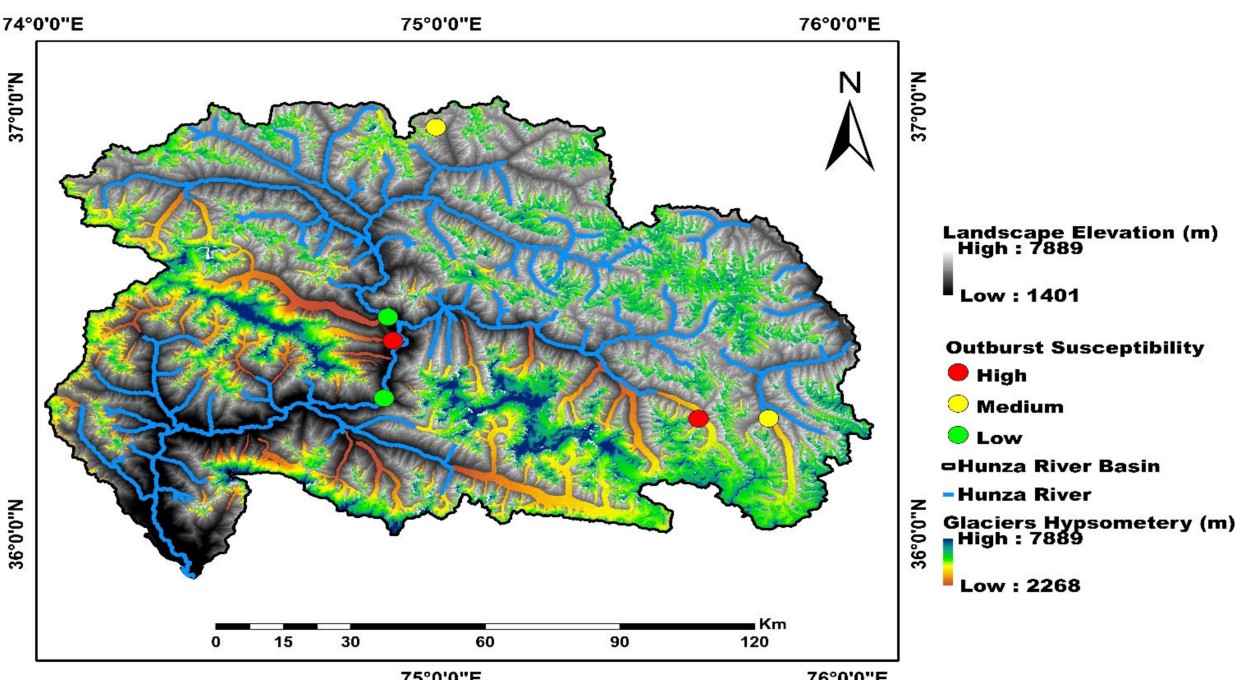

**Figure 8.** Newly formed GLOFs in the Hunza River Basin.

**Table 4.** Criteria weights and rank of parameters calculated using AHP.

| Parameters | Criteria | Rank |
|---|---|---|
| Extreme meteorological events | 0.21 | 1 |
| Lake growth | 0.19 | 2 |
| Freeboard level | 0.10 | 3 |
| Distance from glacier | 0.10 | 4 |
| Slope | 0.08 | 5 |
| Drainage | 0.07 | 6 |
| Area | 0.06 | 7 |
| Volume | 0.06 | 8 |
| Moraine width to height ratio | 0.05 | 9 |
| Type of lake | 0.04 | 10 |
| Aspect | 0.03 | 11 |

The consistency ratio was found to be 0.06, which is less than 1 and indicating good consistency in judgements and is acceptable. The ultimate weight of the individual lake was computed by multiplying parameter weight with $C_i$ of each parameter. The characteristics of the lake such as area, dam type, moraine dam characteristics, glacier lake distance, and meteorological conditions were computed for all lakes. Lastly, outburst susceptibility score was calculated for each lake (Table 5). The outbursts susceptibility was categorized into 3 classes i.e., high, medium, and low on the basis of lake susceptibility score. To verify the anticipated method, we employed the outbursts susceptibility assessment to previous GLOF events in the Hunza Basin. Out of two GLOFs, Passu Lake has an outbursts susceptibility score of 0.69 and Shisper Lake has a score of 0.94 (Table 5). Thus, a susceptibility score of >0.75 was utilized as a starting point for categorizing lakes having high susceptibility, followed by 0.60 demonstrating medium, and 0.50 indicating low outbursts susceptibility.

Analysis indicates that lakes with high susceptibilities are located in proximity with the parent glacier and are therefore exposed to mass movements and ice calving. MCDA reveals a moraine dammed lake, named as Passu lake (Lake-30), is highly susceptible to GLOF in the Hunza River Basin. The Passu Lake was identified as high potential GLOF by the Pakistan Meteorological Department in 2015. Among six identified potential GLOFs, 2 lakes expanded exponentially by 160.7% and 269.59%. The other 2 lakes expanded moderately by 96.18% and

88.07%, while for one lake, the expansion was insignificant, and it was decreased by 10.51% (Table 5). The other lakes categorized as medium to low susceptibilities are not expanding drastically and are characterized by greater freeboard level. The majority of glacial lakes experienced significant growth changes over the decade as shown in Figure 11. On the basis of outbursts probability and prioritization scheme, lake 30 and lake 53 are categorized as having high susceptibility, lake 57 and lake 220 are assigned with medium, and lake 31 and lake 273 with low outbursts susceptibility (Figure 8).

**Table 5.** Characteristics of lakes for GLOF outbursts susceptibility assessment.

| Lake ID | Area (km²) | Volume (m³) | Type | Freeboard (m) | Dam width to Height Ratio | Aspect | Drainage Type | Distance from Glacier (m) | Slope (°) | Lake Growth 2008–2018 (%) | Extreme Meteorological Events | Final Score |
|---|---|---|---|---|---|---|---|---|---|---|---|---|
| Lake 30 | 0.13 | 1,914,435.99 | M | 0 | 0.25 | E | D | 25.3 | 5.34 | 96.18 | High | 0.78 |
| Lake 31 | 0.03 | 237,955.26 | B | NA | NA | E | C | 284.5 | 5.74 | 23.98 | High | 0.50 |
| Lake 53 | 0.03 | 252,724.20 | M | 1.21 | 1.6 | S | D | 29.78 | 5.72 | 160.72 | High | 0.83 |
| Lake 57 | 0.03 | 259,453.24 | I | NA | NA | SW | C | 0 | 6.25 | 269.59 | High | 0.72 |
| Lake 220 | 0.05 | 620,096.11 | M | 8.23 | 3 | E | C | 2505.6 | 9.87 | 88.07 | High | 0.62 |
| Lake 273 | 5.1 | 340,301,839 | O | NA | NA | W | D | 3612.5 | 14.73 | −10.51 | High | 0.51 |
| Outburst susceptibility assessment of Past GLOF events | | | | | | | | | | | | |
| Shisper Lake | 0.3 | 6,501,922 | I | NA | NA | S | C | 0 | 22.15 | 124.84 | High | 0.94 |
| Passu Lake | 0.16 | 2,588,878 | M | 0 | 0.16 | E | D | 11.8 | 5.34 | 4.88 | High | 0.69 |

## 5. Discussion

### 5.1. Glacial Lake Dynamics and Climate Indicators

Climate indicators such as temperature and precipitation influence the mass balance of glaciers, which in turn increase snowfall and glacier melt discharge heading to growth of glacial lakes. The yearly average temperature and rainfall data from 2003–2018 and 1998–2018 for the study area was obtained and analyzed as shown in Figure 9. The annual mean temperature has indicated an increasing trend of 0.07 °C per year. The annual mean precipitation has shown a rising trend of 0.9 mm per year. If a similar trend of precipitation continues, it would lead to further accelerate glacier surges and slow-down glacier melt and could transform bare land into glacier area.

Lake surface water temperature is a proxy in determining the regional climate variability and is an essential insignia of lake stability [49]. The lake surface water temperatures of identified potential GLOFs revealed an increasing warming trend as depicted in Figure 10. This could be owing to the decline in ice cover and glacier retreat. The lake surface water temperature (LSWT) of potential GLOFs may predict that identified potential GLOFs would have outbursts in the future if the LSWT increases with same pattern.

A glacial lake in proximity with glacier has the highest expansion rate, showing glacier melt water as the predominant basis for lake expansion. Glacier lake distance plays pivotal role in glacier dynamics and is an essential precursor to a GLOF event. Glacier lake distance from glaciers observed in this study is an indication of glacier recession by means of feedback mechanism and anticipates lakes expansion in the basin.

Hence, growing in areal extent and number of glacial lakes can be attributed to the enhanced ablation and glacier retreat owing to global warming. Therefore, it can be speculated that with unabated climate change and future warming, glacier recession might rise to further evolution and growth of glacial lakes with greater outbursts vulnerability and GLOF hazards in the Hunza River Basin.

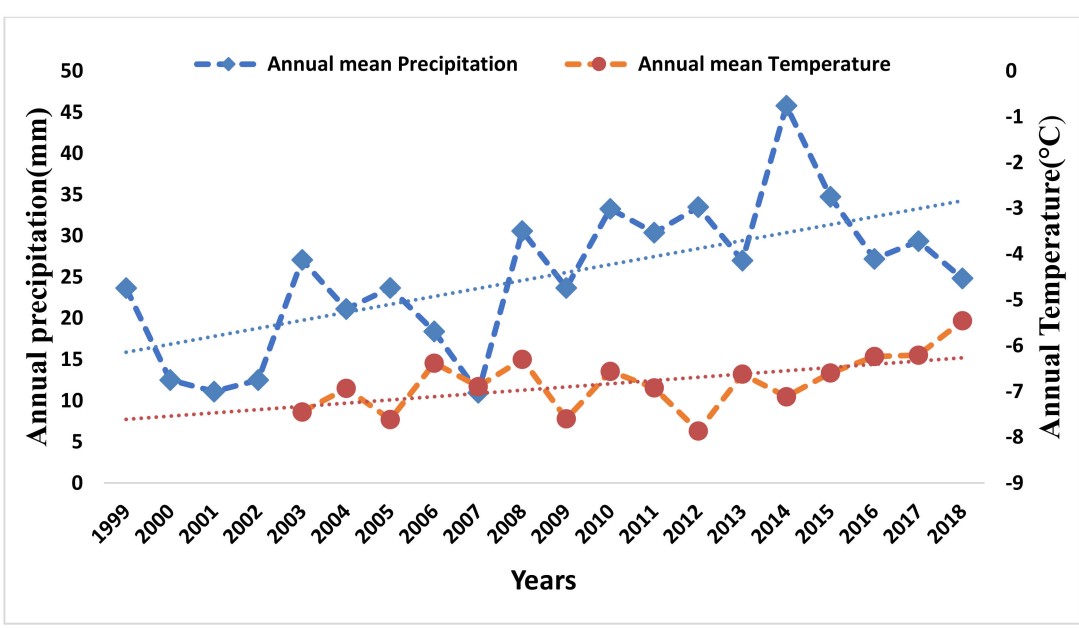

**Figure 9.** Temperature and precipitation change in the Hunza River Basin.

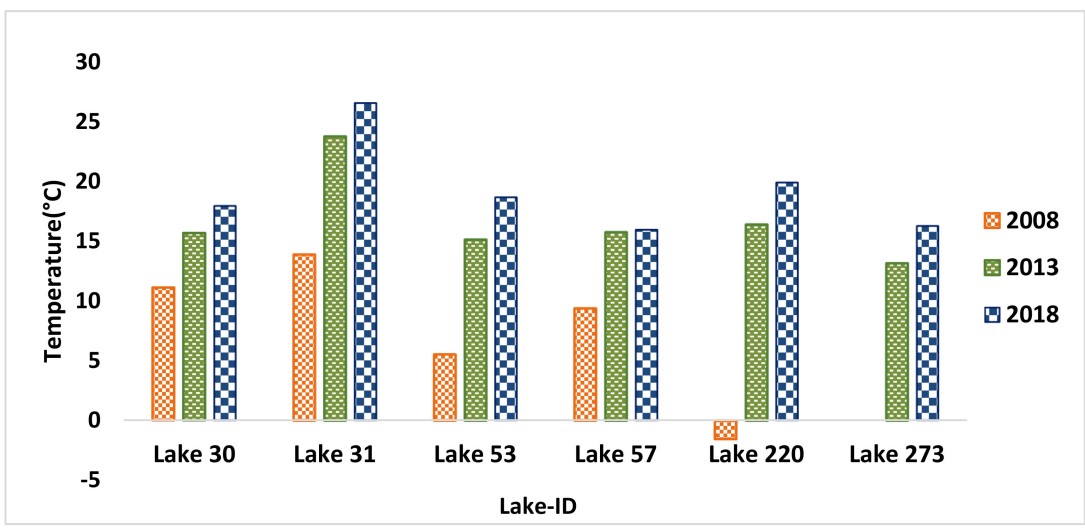

**Figure 10.** Lake surface water temperature of potential GLOFs.

### 5.2. Lakes Prone to Catastrophic Outbursts Flood

Glacial lake outbursts flooding along debris flow is one of the imminent hazards in the basin. An end moraine dammed lake (Passu Lake) with an area of 0.16 km$^2$ closed to Passu glacier had caused glacial lake outbursts flooding in July 2007. The lake had caused damage to hotels, infrastructure, Karakorum Highway, and houses in Passu village [50]. This lake has been again identified as a high potential GLOF in current study. Hence, the lake requires continuous monitoring and inspection on regular basis in order to avoid drastic losses in socioeconomic circles. The lake is highly perilous for residents of Passu village but another lake formed in February 2010 known as Attabad Lake, the villages along the Hunza River became more susceptible for catastrophic outbursts flooding. Shisper Lake formed with an area of 0.13 km$^2$ in December 2018 and had increased to an area of 0.3 km$^2$ in May 2019. The Shisper Lake had a damaged water channel, Karakorum highway, and hydropower station in Hassan Abad [51]. The above-mentioned events revealed that the study area is highly prone to catastrophic glacial lake outbursts flooding, and our approach is effective in identifying potential dangerous glacial lakes.

Figure 11 is showing lake growth of potential GLOFs. All lakes have an area greater than 0.02 km$^2$, and the majority of lakes are connected with the parent glacier. The closer vicinity with the mother glacier not only supplies large amounts of glacier melt water, but also offers more chances of rock falls or ice avalanches entering into the lake, which are essential precursors of triggering outbursts of glacial lakes because about 75% of GLOF events have been instigated by ice avalanche [35]. Out of six, 5 lakes experienced area expansion of 23.98% to 269.59%, and one lake showed area reduction, though it has been appraised as low outburst potential and still requires monitoring. Some of glacial lakes have close proximity with the parent glacier, gentle slopes of lake suggest space for areal expansion as illustrated in Figure 11. Hence, glacial lakes need attention to comprehend the dynamics and mechanism of these lakes in the future, and ground observations are required. Nevertheless, the study offers forecast of possible risk of GLOF events in the region owing to unabated climate change and increasing global warming worldwide. Once the glacial lake outbursts failure occurs, downstream communities, property, fragile ecosystems, infrastructure would be affected and damaged same as done by earlier GLOF disasters in the HRB.

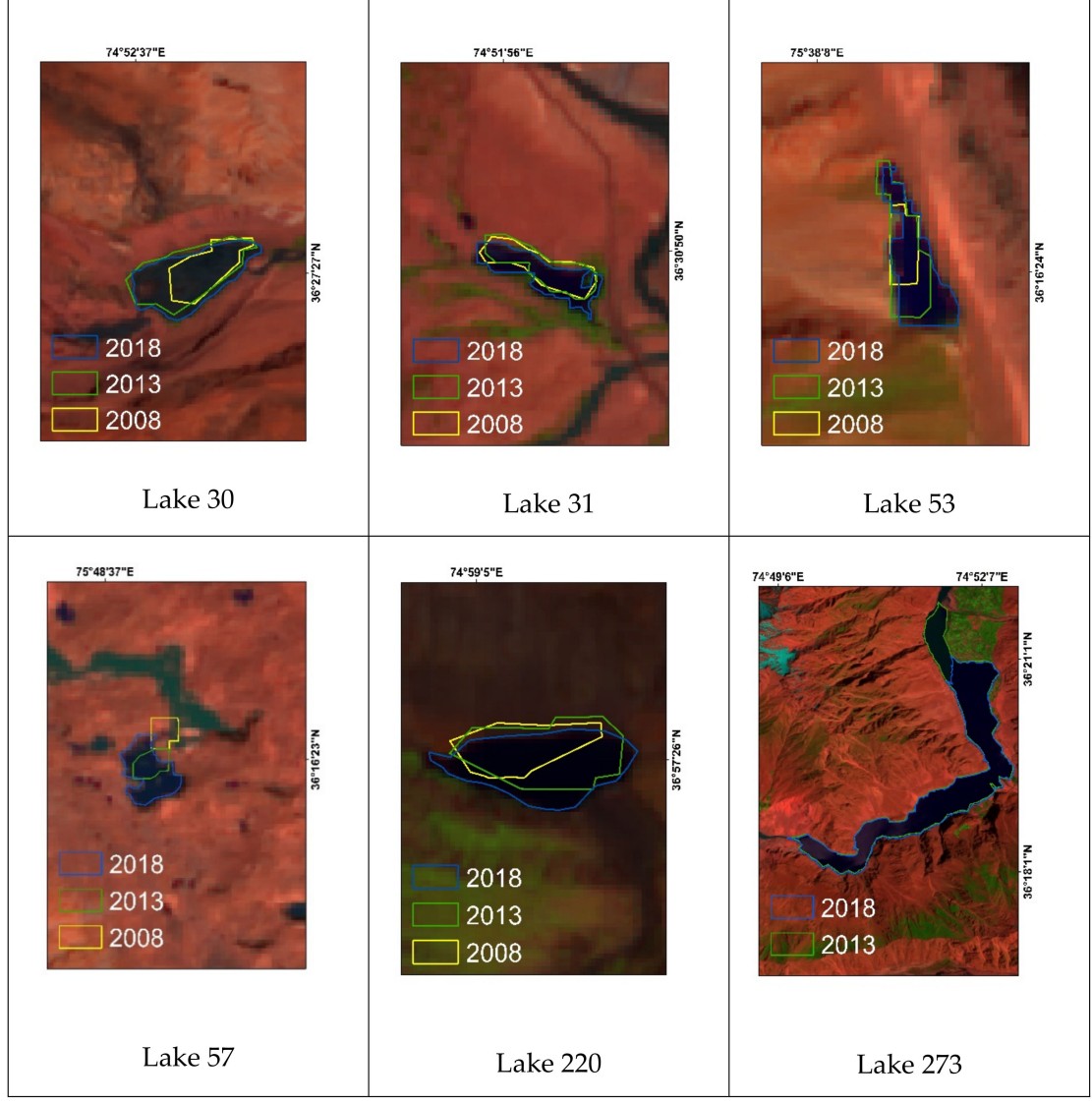

**Figure 11.** Illustration of lake growth of potential GLOFs.

*5.3. GLOF Susceptibility Assessment*

The aforementioned results exhibit that lakes detection using NDWI leads to accurate outcomes. This also coincides with earlier researches [52]. In lake identification, manual improvements synchronized with visual interpretation is considered to be a prerequisite. The automated lake identification can be difficult and challenging owing to the presence of partial ice cover, debris, shadow regions, and turbidity in glacial lakes [53]. However, this study adopted a semi-automated approach to identify glacial lakes, which is in congruence with past studies, e.g., Begam et al. [54]; Prakash and Nagarajan [55].

The current study has employed a first order method to detect and prioritize potential hazardous glacial lakes. GLOF outbursts susceptibility was established by means of satellite data, GIS tools, and AHP. This method has been successfully utilized to identify potential dangerous lakes by Yiong Zango Basin, Chandra Bhaga Basin, and Sikkim Basin [12,32,55]. We have selected 11 factors based on trigger mechanisms and conditions of past GLOF events as documented in literature. These factors indicate status of glacial lake (area, volume, dam type, aspect, and drainage condition, lake growth, mean slope of the lake), moraine dam (free board level, dam with to height ratio), glacier-lake dynamics (distance between glacier and glacial lake), and extreme meteorological conditions. We are of the opinion that these factors can fulfill requirements for risk assessment of glacial lakes. Some factors such as geomorphology of moraine dams are not taken into account due to lack of high-resolution data such as presence of internal ice underlying the moraine dam. However, the present study is based on medium resolution Sentinel-2 imageries and provides good details about area, type, and geomatics of the lake. Moreover, glacial lakes are dynamic in nature, those having low outbursts potential can evolve into high outbursts potential owing to fluctuations in areal extent of glacial lakes or glaciers surging and recession.

The AHP-based approach for outbursts susceptibility has more advantages than other methods because it is simple, versatile, and mathematical, employs qualitative and quantitative evaluation of parameters by researcher or experts in GLOF assessment, and can hierarchize many factors affecting complex problems and rank them based on experts' judgement. However, this method has certain limitations, such as it employs manual weighing scheme and involves various experts to ensure consistency in judgments through pairwise comparison. Despite the few limitations, the method employed for this study is efficient and effective for first order evaluation of outbursts susceptibility of potential glacial lakes at the regional scale for explicit investigations in the Karakorum region.

## 6. Conclusions

Climatic alterations and concomitant glacier recession have led to the evolution, growth, and development of glacial lakes in high mountain areas worldwide. Comprehensive glacial lake inventory has been established for the Hunza River Basin using satellite remote sensing data and subsequently analyzed glacial lakes with respect to their GLOF susceptibility. In terms of glacial lake type, about 52.3% (n = 154) of the total glacial lakes were impounded by ice dams with mean elevation of 3871 m whilst 39.7% (n = 39) glacial lakes were impounded by moraine dams situated at mean elevation of 4298 m followed by bedrock lakes (5.78%, n = 17). Other glacial lakes (2.04%, n = 6) are located at a mean elevation of 4080 and 2969 m. The glacial lakes exhibited inconsistent distribution characteristics with respective to both number and area; a large number of glacial lakes were distributed above 4500 m, but a greater total area of glacial lakes was situated below 2500 m. About 45 lakes out of 294 glacial lakes having an area >0.01 km$^2$ were assessed using predefined criteria for GLOF susceptibility assessment, and six lakes were recognized as potential GLOF. Commencing the investigation of previous GLOF events, we observed that events have been caused by glacier recession or advancement and triggered by the temperature. The time of upsurges begin from June to September and is characterized by summer monsoon in the area heading to enhance water level in the lakes. Hence, mass movements and climatic settings play a pivotal role in GLOF events. Past GLOF events in the Hunza River Basin were utilized to define the GLOF susceptibility factors of glacial lakes using AHP. The

approach categorized two lakes as highly susceptible, two lakes are classified as medium, and two lakes are assigned with low outbursts susceptibility. Glacial lake outburst hazard is anticipated to rise in the Hunza River Basin owing to escalating global warming and glacier fluctuations. These lakes should be monitored continuously, and profound modelling, potential flood volume estimations, and simulations for GLOFs need to be addressed in detail on the basis of in situ observations and field measurements for GLOF risk mitigation and hazard management across the Hunza River Basin. This study presents an up-to-date and comprehensive inventory of glacial lakes and their susceptibility to GLOF prerequisites for planning and risk assessment considering socio-economic dynamics in the Hunza River Basin. This study might be helpful for various institutions in preparing better adaptation strategies, particularly for those identified as having a high potential of outbursts flooding. It might assist in sustainable development of fragile mountainous regions in setting up early warning systems at potential GLOF regions.

**Author Contributions:** Conceptualization, F.M., M.F.K. and S.U.B.; methodology, F.M., J.A.K.; formal analysis, M.F.K.; writing—original draft preparation, F.M.; writing—review and editing, FM., S.U.B., M.F.K., and J.A.K.; visualization, F.M.; supervision, M.F.K. All authors have read and agreed to the published version of the manuscript.

**Funding:** Authors are grateful to R&D Funds for Post-Graduate from NUST, Pakistan to conduct this study.

**Acknowledgments:** The authors acknowledge the ICIMOD for providing glacial lake inventory data, the NASA USGS and European Copernicus team for providing satellite data. Authors are also thankful to Glacier Lake Ice Measurement from Space (GLIMS) for providing Randolph Glacier Inventory data.

**Conflicts of Interest:** The authors declare no conflict of interest.

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
