# Peer review of "Inventory and GLOF Susceptibility of Glacial Lakes in Hunza River Basin, Western Karakorum"

_remotesensing, doi:10.3390/rs13091794_

Round 1

Reviewer 1 Report

Fig. 6a: please specify the unit of the Glacier lake size classes

Fig. 7: specify in the legend the meaning of the classes I, M, B, and O

Reviewer 2 Report

Nicely presented and interesting paper that potentially will be of great interest both nationally and internationally!

Have suggested a few minor changes, shown as 'sticky notes' in the attached pdf file of the manuscript.

Reviewer 3 Report

This study provides an inventory of glacial lakes in 2018 in Hunza River Basin, Western Karakorum. The authors also evaluated GLOF susceptibility by AHP method. This manuscript is suitable to publish in Remote Sensing, with some improvements of comments below.

Major comments:

  1. How was the glacial lake volume estimated? Why the authors select this empirical formula? The authors compared with others or validated by measured bathymetry?
  2. The inventory of glacial lakes in 2018 is only provided. The more glacial lake mapping in early stage should be provided, so the changes in glacial lake area could be added in aiding to detect potentially dangerous lakes or combining with AHP method.
  3. All the figures need improve greatly to meet the criterion of publication.

Specific comments:

- “Northern latitudes of Pakistan are warming at faster rate as compared to the rest of the country.” This sentence could be removed as it is not special.

- “1.04°C from 1960 to 2014” Please add the rate here

- “during 1900-2000” The data used should be indicated.

- “at the rate of 1.79°C” Please add the change rate here

- “44.02% in 1989 to 34.99% in 2010” revise to change here

- “In high mountain regions where ground data collection and field observations are obstructed by harsh weather and climatic conditions, remote sensing techniques offer flexible approach for spatial and temporal assessment and monitoring of glacial lakes and potential GLOFs [19]–[21]. ” Suggested reference here: doi: 10.1017/jog.2019.13

- “AHP based multi-criteria has been widely employed in other regions for outbursts susceptibility assessment[27]–[30].” Suggested reference here: doi:10.3389/feart.2020.601288

- “10 x 106” to “10 x 10^6” Others are similar.

- “susceptibility of susceptible glacial”? Change this sentence

- Figures: Sq.Km to km^2, Meters to m. Figures 4, 5,8 should be improved. It is difficult to read as the background image is too shine.  Figures 6,7,9,10 should be improved greatly as it did not meet the criterion of publication.  

- Too many tables. Some can be combined together or put in the supporting information.

Round 2

Reviewer 3 Report

I agree with the publication of this manuscript. It would be better if the authors can improve some figures such as Figures 6, 7, 9, 10 (use professonal software instead of Excel directly).